# A Mixture of Dietary Plant Sterols at Nutritional Relevant Serum Concentration Inhibits Extrinsic Pathway of Eryptosis Induced by Cigarette Smoke Extract

**DOI:** 10.3390/ijms24021264

**Published:** 2023-01-09

**Authors:** Ignazio Restivo, Alessandro Attanzio, Luisa Tesoriere, Mario Allegra, Guadalupe Garcia-Llatas, Antonio Cilla

**Affiliations:** 1Department of Biological, Chemical and Pharmaceutical Sciences and Technologies, University of Palermo, Via Archirafi 28, 90123 Palermo, Italy; 2Nutrition and Food Science Area, Faculty of Pharmacy, University of Valencia, Avda. Vicente Andrés Estellés s/n, 46100 Burjassot, Spain

**Keywords:** cigarette smoke, plant sterols, eryptosis, DISC, caspases, p38 MAPK

## Abstract

Cell death program of red blood cells (RBCs), called eryptosis, is characterized by activation of caspases and scrambling of membrane phospholipids with externalization of phosphatidylserine (PS). Excessive eryptosis confers a procoagulant phenotype and is implicated in impairment of microcirculation and increased prothrombotic risk. It has recently been reported that cigarette smokers have high levels of circulating eryptotic erythrocytes, and a possible contribution of eryptosis to the vaso-occlusive complications associated to cigarette smoke has been postulated. In this study, we demonstrate how a mixture of plant sterols (MPtS) consisting of β-sitosterol, campesterol and stigmasterol, at serum concentration reached after ingestion of a drink enriched with plant sterols, inhibits eryptosis induced by cigarette smoke extract (CSE). Isolated RBCs were exposed for 4 h to CSE (10–20% *v*/*v*). When RBCs were co-treated with CSE in the presence of 22 µM MPtS, a significant reduction of the measured hallmarks of apoptotic death like assembly of the death-inducing signaling complex (DISC), PS outsourced, ceramide production, cleaved forms of caspase 8/caspase 3, and phosphorylated p38 MAPK, was evident. The new beneficial properties of plant sterols on CSE-induced eryptosis presented in this work open new perspectives to prevent the negative physio-pathological events caused by the eryptotic red blood cells circulating in smokers.

## 1. Introduction

“Tobacco is one of the greatest emerging health disasters in the history of humanity”. That is the sentence of Dr. Gro Brandtland, then Director General of the World Health Organization (WHO), in 1998. In 2019, WHO sources say that up to half of tobacco consumers, in all its forms, face death, and that every year, due to the direct use of tobacco or passive smoking, there are about 8 million deaths. Although tobacco continues to be a dramatic factor in the development of cardiovascular diseases and dysfunctions, which are the greatest cause of death in humans, in about twenty years, global tobacco consumption has fallen by only 4.3% (1397 billion consumers to 1337) [1].

A number of studies correlate cigarette smoke (CS) with dysfunction of endothelial cells [2], leukocytes [3], and platelet activation [4], as well as increased circulation of eryptotic RBCs [5] leading to damage of the cardiovascular system. Endothelial dysfunction is one of the main early pathophysiological biomarkers of smokers [6], and relatively recent studies show that excessive and dysfunctional eryptosis, or programmed death of red blood cells (RBCs), is associated with vase-occlusive events and physio-pathogenesis of the cardiovascular system such as atherosclerosis [7] and atherothrombosis [8,9], as well as being involved in chronic kidney disease [10] and metabolic syndrome [11]. Eryptosis is one of the main ways of death of RBCs, and is characterized by loss of membrane asymmetry following scrambling of phospholipids with externalization of phosphatidylserine (PS), membrane shrinkage, membrane blebbing, and activation of proteases involved in cell death such as caspase 8 and caspase 3 (Figure 1) [12].

In a previous study of our research groups, we have shown that smokers have a higher level of circulating eryptotic erythrocytes associated with inflammatory status (high C-reactive protein levels) and RBCs’ oxidative stress (low glutathione levels), suggesting a further relationship between cigarette smoke and vascular damage [5]. In this regard, we recently investigated, in isolated human RBCs, the molecular mechanism of the eryptosis induced by cigarette smoke extract (CSE). The study showed that CSE induces extrinsic eryptosis where assembly of the death-inducing signaling complex (DISC) is started by activating p38 MAPK, and followed by overproduction of ceramide and activation of caspase 8 and caspase 3. In vitro results were reinforced by ex vivo data where RBCs from smokers report higher levels of caspase 8 and FADD in the DISC, as well as increased phosphorylation of p38 MAPK than non-smokers [13].

According to the statistical office of the European Union (EUROSTAT), cardiovascular disease continues to be the main cause of death for humans [14], and it is a main objective of WHO to reduce deaths from these diseases [1]. Therefore, it is very important to implement prevention and try to inhibit all possible causes of cardiovascular disease, including eryptosis. Although efforts have been made to reduce the number of global smokers in order to reduce care costs and mortality, in recent years, the number of smokers has fallen by a few million. It is therefore very important and cost-effective to use nutrition to help reduce the onset of diseases, but above all, as a preventive tool.

Plant sterols (PtS) are important phytochemicals found in plant foods, and the most common present in foods are sitosterol, campesterol, and stigmasterol, which account for about 50–65%, 10–40%, and 0–35% of the total PtS fraction. PtS are well known for their positive health activities, so their beneficial role has been increasingly consolidated over the years [15,16]. PtS, naturally occurring mainly in vegetable oils, margarines, and nuts, are steroids associated with health benefits such as cholesterol-lowering, anti-inflammatory, and anti-proliferative effects [17]. PtS are normally absorbed with low efficiency, and different PtS-enriched foods are now commercialized to obtain the recommended daily quantity for healthy purposes. Previous study showed that four months of consumption of milk-based fruit beverage enriched with 1.5 g of mixture of plant sterols (MPtS) consisting of β-sitosterol, campesterol, and stigmasterol (Figure 2), resulted in an increase in serum concentration of plant sterols up to 22 μM [18]. At nutritional relevant serum concentration, MPtS was shown to prevent pro-eryptotic and hemolytic effects evoked by oxidative stress induced by tert-butyl hydroperoxide (tBOOH) in isolated human RBCs [19].

The aim of this work is investigating, for the first time, the potential protective effect of MPtS, at serum concentration reached after the consumption of PtS-enriched milk-based fruit beverage, against eryptosis induced by CSE.

## 2. Results

### 2.1. MPtS Decreases PS Exposure and Ceramide Production in CSE-Induced Eryptosis

Since CSE-induced extrinsic eryptosis involves DISC in the first hours of treatment [10], 0.4% hematocrit of isolated RBCs were treated for 4 h with 10% CSE and 20% CSE (*v*/*v*), with the absence of treatment or after combination with 22 μM MPtS. Parameters characterizing eryptosis such as membrane scrambling with PS exposure, ceramide production, and cell volume variation measured by forward scatter (FS), were evaluated by flow cytometry. Co-treatment of erythrocytes with 22 μM MPtS, decreased (*p* < 0.001) the CSE-induced increase of annexin V-binding fluorescent cells with 20% CSE treatment (5.9% vs. 11%) and 10% CSE treatment (5.8% vs. 8.8%), reporting values on control levels (Figure 3A). Treatment of RBCs with only MPtS did not show any increase of PS exposure (4.7% vs. 4.6%). Likewise, ceramide formation was significantly (*p* < 0.001) inhibited by MPtS, as shown by the decrease of cells binding anti-ceramide labeled with fluorescein-5-isothiocyanate (FITC) probe with 20% CSE treatment (4.4% vs. 13.5%) and 10% CSE treatment (4.4% vs. 9.2%) (Figure 3B). In addition, FS measurements failed to pick a variation of cell volume with any treatment (Appendix A).

### 2.2. MPtS Inhibits CSE-Induced Extrinsic Eryptosis

To assess whether MPtS inhibited CSE-induced extrinsic eryptosis, co-immunoprecipitation experiments of RBCs treated with either 10% or 20% CSE in the absence or in the presence of 22 μM MPtS, were performed to assess formation of DISC in the membrane. After 4 h of treatment or co-treatment, the RBCs’ lysates were immunoprecipitated with anti-FAS antibodies, followed by Western blotting with anti-FAS-associated via death-domain (FADD) or anti-caspase 8. Both caspase 8 and FADD were found in the immunoprecipitates of RBCs treated with CSE, in a concentration-dependent manner (Figure 4). Co-treatment with MPtS, on the other hand, showed an evident decrease almost to reach control levels (*p* < 0.001) in the amounts of both proteins analyzed with all the conditions of treatment (Figure 4B,C). The RBCs treated only with MPtS did not show any significant changes in the proteins’ asset.

### 2.3. MPtS Inhibits Caspase 8/Caspase 3 Cleavage and Phosphorylation of p38 MAPK in CSE-Induced Extrinsic Eryptosis

After treating the RBCs for 4 h, the cell lysates were immunoblotted with anti-caspase 8 or anti-caspase 3. When the RBCs were co-incubated with 22 μM MPtS, there was a significant reduction (*p* < 0.001) in the level of the active fragments of both caspase 8 and caspase 3 (Figure 5C,D), compared to cells treated only with CSE which showed a much higher level of the cleaved forms. Activation and signaling of the p38 MAP kinase pathway also appears to be involved in several eryptotic pathways and is characterized by the phosphorylation of the enzyme. Co-treatment at 4 h with MPtS showed an evident reduction (*p* < 0.001) in the phosphorylation of p38 MAPK, both with 10% CSE and 20% CSE treatment, compared to RBC only treated with CSE at both the concentrations (Figure 5B). RBCs treated only with MPtS did not report significant changes in the proteins analyzed above.

## 3. Discussion

In the present study, we have shown that MPtS at 22 μM, used at this concentration as compatible with nutritional relevant serum concentrations obtained after the consumption of PtS-enriched milk-based fruit beverage [18], inhibits CSE-induced extrinsic eryptosis. MPtS counteracts eryptosis with the reduction of the pro-eryptotic bioactive lipid ceramide, the caspase 8 and caspase 3 proteases activation, the formation of the DISC in membrane and the phosphorylation of p38 (p-p38) MAPK. All these aspects lead to a reduction of the externalization of PS which is the key element of the eryptotic RBCs to be recognized by macrophages and to be removed from the bloodstream [20,21,22].

Although they represent hallmarks characteristic of the eryptotic process [8,9] changes in redox balance, intracellular Ca^2+^ levels and cell volume measured by FS, have not been observed (Appendix A). Present data confirm what has been observed previously [13]. However, the lack of variation of the FS could be partly attributed to the time of treatment in our model, since 4 h of treatment under experimental conditions could not be sufficient to unbalance the cellular volume. In recent work investigating the biochemical pathway leading to eryptosis in RBC treated with CSE [13] and in line with other studies [23,24], we have shown that the stimulation of p38 MAPK leads to the assembly of the membrane DISC resulting in caspase activation. The mechanism in which this happens is unclear and deserves to be explored. Our data suggest that MPtS is able to inhibit the eryptotic process by inhibiting p-p38 MAPK which, as mentioned above, is the initiator of CSE-induced extrinsic eryptosis. In addition, the chemical-physical properties of PtS, could make them interact with the cell membrane of RBC going in some way to strengthen the inhibition of eryptosis avoiding the trimerization of FAS acting on membrane lipids-raft. It is quoted that MPtS can interact with the pathway involving Apoptosis Signal-Regulating Kinase 1 (ASK1)- MAPK kinases (MKK) 3/6 axis that in different experimental models leads to the phosphorylation and activation of p38 MAPK [25,26,27]. These assumptions would be in line with Zhang et al. [28] and Xu et al. [29] where the PtS like β-sitosterol and ergosterol appear to have modulated activities of p38 MAPK in gastric epithelial GES-1 and murine macrophages RAW 264.7 cells. However, to the best of our knowledge, this work is the first study in which it has been shown the inhibition of eryptosis by plant sterols acting on the extrinsic death pathway. For nucleated cells, many biochemical mechanisms are well known, while for anucleate cells such as RBCs, many mechanisms remain unknown or poorly studied. This study provides many ideas of reflection and deepening to fill the gap in the pathways shown above.

In conclusion, there are many pharmacological approaches and awareness campaigns aimed to reducing the harm of smoking. The new beneficial properties on RBC presented in this work involving plant sterols at nutritional relevant serum concentrations compatible with their well-known hypocholesterolemic effects, open new perspectives to prevent and treat part of the negative physio-pathological events caused by the eryptotic red blood cells circulating in smokers. Further confirmatory in vivo studies are warranted.

## 4. Materials and Methods

When not specified, all chemicals were purchased from Sigma-Aldrich (Milan, Italy) and were of the highest purity grade available.

### 4.1. Preparation of CSE

CSE was prepared by a modification of a methods previously published by Carnevali et al. [30]. In summary, 3 filtered Marlboro Red cigarettes (Philip Morris USA Inc., Richmond, VA, USA), each containing 0.8 mg nicotine, 10 mg tar and 10 mg carbon monoxide, were smoked consecutively through a system with a constant airflow (0.4 L/min) controlled by a vacuum pump. The smoke was gurgling through 30 mL of Ringer solution pre-heated to 37 °C. The Ringer solution consist of (mM) 125 NaCl, 5 KCl, 1 MgSO_4_, 32 N-2-hydroxyethylpiperazine-N-2-ethanesulfonic acid (HEPES)/NaOH, 5 glucose, 1 CaCl_2_, with a final pH 7.4. Cigarettes were torched up to 3 mm from the start of the filter. To ensure its sterility CSE was filtered through a 0.22-mm filter (Millipore, Billerica, MA, USA). This solution was considered to be 100% CSE. Concentration of nicotine, one of the stable constituents, was assessed by LC–MS analysis carried out using an Ultimate 3000 instrument coupled to a TSQ Quantiva (Thermo Fisher Scientific, San José, CA, USA) triple-stage quadrupole mass spectrometer. Nicotine mean concentration of four different CSE preparations was 32.13 ± 1.23 mg/L.

### 4.2. Preparation of MPtS

The PtS mixture at 22 μM final concentration in Ringer solution compatible with nutritional relevant serum concentrations obtained after the consumption of PtS-enriched milk-based fruit beverage, cointain β-sitosterol (13 μM), campesterol (8 μM), and stigmasterol (1 μM) and was prepared at 0.2% (*v*/*v*) ethanol, as reported in a previous study [18].

### 4.3. Red Blood Cells and Treatment

Fresh blood samples were collected in tubes coated with lithium heparin or sodium heparin to inhibit clotting from non-smoking healthy male volunteers (*n* = 6; age range 25–51 years; normal BMI range) with informed consent, and RBCs were immediately isolated by centrifugation (2000× *g*, 4 °C, 20 min) over a Ficoll (Sigma-Aldrich, Milan, Italy, Cat. No. F5415) gradient [31]. Cells were washed twice in Ringer solution. The cell pellet aliquots were diluted in order to have a 0.4% hematocrit (HT) in the same simple Ringer solution (control) or in Ringer containing CSE at appropriate dilutions for the treatment. RBCs were then incubated at 37 °C, 5% CO_2_ and 95% humidity for 4 h.

The experimental study protocol was approved by the Ethic Committee of Palermo 1, University Hospital (No. 8-09/2022) and performed in accordance with the Declaration of Helsinki and its amendments.

### 4.4. Measurement of PS Externalization

RBCs were washed once in Ringer solution, pH 7.4, and adjusted to 1.0 × 10^6^ cells/mL with binding buffer following the manufacturer’s instructions (eBioscience, San Diego, CA, USA, Cat No. 88-8005-74). In experiments designed to evaluate the percentage of PS externalization, suspension of RBCs (100 µL) was incubated with 5 µL of Annexin V-FITC at room temperature in the dark for 15 min. Subsequently, suspension samples of at least 1.0 × 10^4^ cells were subjected to flow cytometric analysis by Epics XL™, using Expo32 software (Beckman Coulter, Fullerton, CA, USA). The annexin V-fluorescence intensity was measured in fluorescence channel FL-1 with an excitation wavelength of 488 nm and an emission wavelength of 530 nm.

### 4.5. Measurement of Ceramide

Abundance of ceramide levels was measured as follows. Briefly, after treatment, 1.0 × 10^5^ erythrocytes were incubated for 1 h at 37 °C with 1 µg/mL of a mouse monoclonal anti-ceramide antibody (Sigma-Aldrich Chemical Co., St. Louis, MO, USA, Cat. No. C8104) in PBS containing 0.1% bovine serum albumin (BSA). RBCs after two washing steps with PBS-BSA, were stained for 30 min with 20 µL of a goat anti-mouse, polyclonal, fluorescein isothiocyanate-conjugated, secondary antibody (Millipore, Billerica, MA, USA, Cat. No. AQ502F) diluted 1/50 in PBS-BSA in the dark. Finally, erythrocytes were collected by centrifugation (2000× *g*, 4 °C, 5 min), washed twice, resuspended in PBS and analyzed by flow cytometer as reported in Section 4.4.

### 4.6. Immunoprecipitation

RBCs (2.0 × 10^8^ cells) were washed twice with PBS and resuspended in lysis buffer (20 mM Tris-HCl, pH 7.6, 100 mM NaCl, 10 mM MgCl_2_, 1% NP-40, 2 mM PMSF, 0.5 mM DTT and 2 mg/mL lysozyme) containing phosphatase (Roche, Basel, Switzerland, Cat. No. 4906845001) and protease inhibitors (Roche, Basel, Switzerland, Cat. No. 4693132001). Whole cell lysates were also sonicated (2 cycles, each for 30 s) with Labsonic LBS1-10 (Labsonic Falc, Treviglio, Italy) and after centrifugation (40,000× *g*, 4 °C, 1 h) were incubated overnight with mouse anti-FAS antibody (1:200) at 4 °C. Supernatants were then incubated with 20 μL of Protein G PLUS-Agarose (Santa Cruz Biotechnology, Inc., Dallas, TX, USA, sc-2002) for 3 h at 4 °C. Beads were pelleted, washed twice in lysis buffer and finally proteins were separated by SDS-PAGE for immunorecognition by western blotting [13].

### 4.7. Western Blotting

After treatment, erythrocytes (2.0 × 10^8^ cells) were washed twice with PBS, resuspended in lysis buffer (20 mM Tris-HCl, pH 7.6, 100 mM NaCl, 10 mM MgCl2, 1% NP-40, 2 mM PMSF, 0.5 mM DTT and 2 mg/mL lysozyme) containing phosphatase and protease inhibitors and sonicated (2 cycles, each for 30 s) with Labsonic LBS1-10 (Labsonic Falc, Treviglio, Italy). Lysates were clarified by centrifugation (40,000× *g*, 4 °C, 1 h) and supernatants were collected and stored at −80 °C. Bradford protein assay (Bio-Rad, Hercules, CA, USA, Cat. No. 5000006) was used to quantify the total protein concentration in each sample. For each sample, equal amounts of proteins were loaded (50 µg/lane), separated on 10% gel by discontinuous SDS-PAGE and then electrotransferred to a polyvinylidene difluoride (PVDF) membrane (Millipore, Billerica, MA, USA, Cat. No. IPVH00010). Blots were treated with blocking solution (5% nonfat dry milk) and then incubated overnight at 4 °C with primary antibodies (Santa Cruz Biotechnology, Inc., Dallas, TX, USA) in Tris-buffered saline (TBS; 25 mM Tris, 150 mM NaCl, pH 7.4) containing Tween 20 (1%, *v*/*v*) (TBST) and 5% (*w*/*v*) BSA. Mouse monoclonal anti-FADD (Santa Cruz Biotechnology, Inc., Dallas, TX, USA, sc-271748), anti-FAS (Santa Cruz Biotechnology, Inc., Dallas, TX, USA, sc-74540), anti-caspase 3 (Santa Cruz Biotechnology, Inc., Dallas, TX, USA, sc-56053), anti-caspase 8 (Santa Cruz Biotechnology, Inc., Dallas, TX, USA, sc-81657), and anti-p-p38 MAPK (Santa Cruz Biotechnology, Inc., Dallas, TX, USA, sc-166182) primary antibodies were used at a dilution of 1:200. After washing three times with TBST, immunoblots were incubated with a 1:2000 dilution of rabbit anti-mouse IgG antibody, horseradish peroxidase (HRP) conjugated (Sigma-Aldrich Chemical Co., St. Louis, MO, USA, Cat. No. AP160P) for 1 h at room temperature. Immunoblots were then washed five times with TBST and developed by enhanced chemiluminescence (Amersham, Milan, Italy, Cat. No. RPN2232). Mouse monoclonal anti-actin antibody (Santa Cruz Biotechnology, Inc., Dallas, TX, USA, sc-8432) was used as loading control. Densitometric analysis of protein spots was measured by Quantity One Imaging Software (Bio-Rad, Hercules, CA, USA, Cat. No. 1708265) and the results were reported as arbitrary densitometric units normalized to actin [32].

### 4.8. Statistical Analysis

Results are expressed as mean ± SD of (*n* = 6) separate experiments in triplicates. Statistical comparisons were performed by one-way analysis of variance (ANOVA) followed by Tukey’s correction for multiple comparisons using Prism 8.4 (GraphPad Software Inc., San Diego, CA, USA). In all cases, significance was accepted if the null hypothesis was rejected at the *p* < 0.05 level.

## Figures and Tables

**Figure 1 ijms-24-01264-f001:**
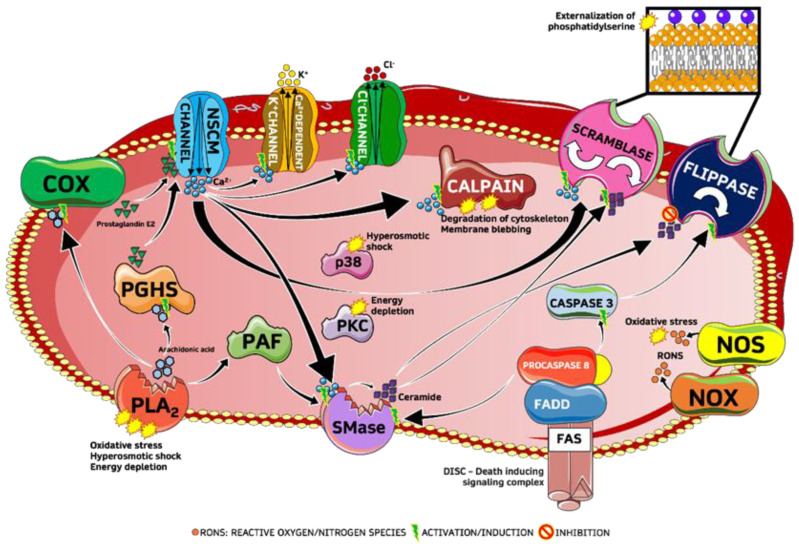
Synthetic scheme of eryptotic machinery. Sphingomyelinase (SMase), platelet-activating factor (PAF), phospholipase A_2_ (PLA_2_), prostaglandin endoperoxide synthase (PGHS), cyclooxygenase (COX), calcium sensitive non-selective (NSCM), protein kinase C (PKC), FAS-associated via death-domain (FADD), nitric oxide synthase (NOS), NADPH oxidase (NOX).

**Figure 2 ijms-24-01264-f002:**
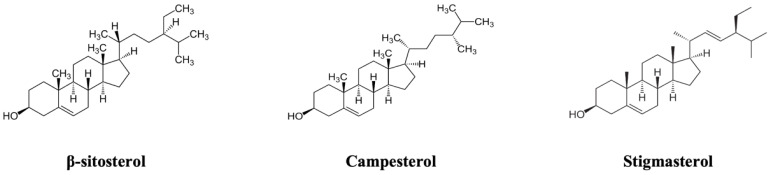
Structure of main dietary plant sterols.

**Figure 3 ijms-24-01264-f003:**
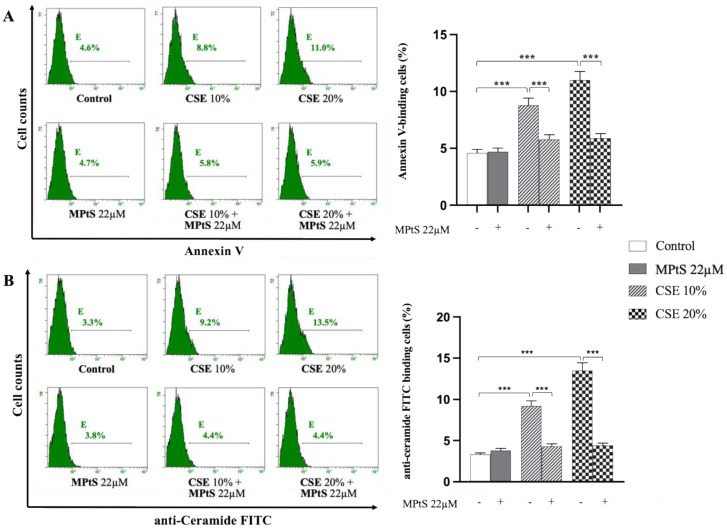
Mixture of plant sterols (MPtS) inhibits cigarette smoke extract (CSE)-induced eryptosis with decrease of PS exposure and ceramide production. (**A**) Percentage of PS-exposing erythrocytes and (**B**) ceramide formation in 10% or 20% CSE-treated RBCs incubated for 4 h in the absence or in co-treatment with 22 μM MPtS measured by flow cytometry. RBCs incubated with Ringer solution were used as control. Protocol of measurement at Section 4.4. Values are means ± SD of (*n* = 6) experiments carried out in triplicate. *** *p* < 0.001 (ANOVA associated with Tukey’s test).

**Figure 4 ijms-24-01264-f004:**
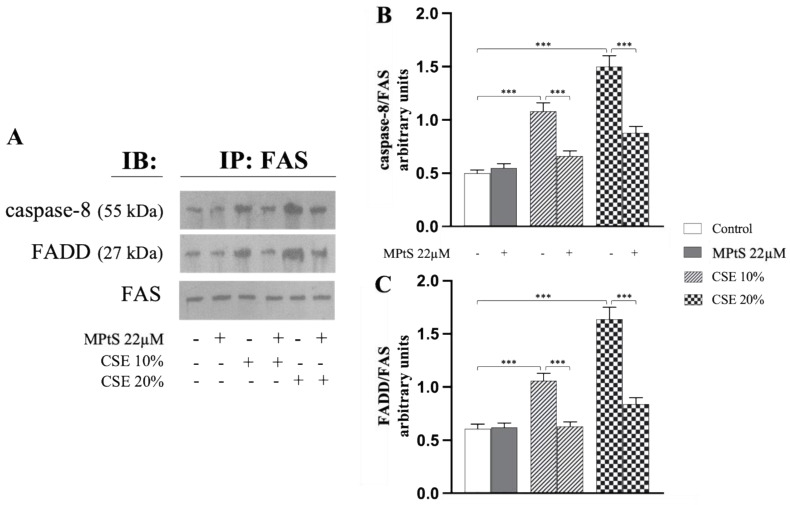
Mixture of plant sterols (MPtS) inhibits cigarette smoke extract (CSE)-induced extrinsic eryptosis. (**A**) Representative images of immunoblotting analysis of caspase 8 and FAS-associated via death-domain (FADD) after immunoprecipitation with anti-FAS antibody in 10% or 20% CSE-treated RBCs incubated for 4 h in absence or in co-treatment with 22 μM MPtS. Protocol of analysis at Section 4.6 and Section 4.7. (**B**,**C**) densitometric analysis of caspase 8 and FADD levels normalized for FAS. Values are the means of bands’ densitometry of three separate experiments with comparable results carried out on the same blood sample. *** *p* < 0.001 (ANOVA associated with Tukey’s test).

**Figure 5 ijms-24-01264-f005:**
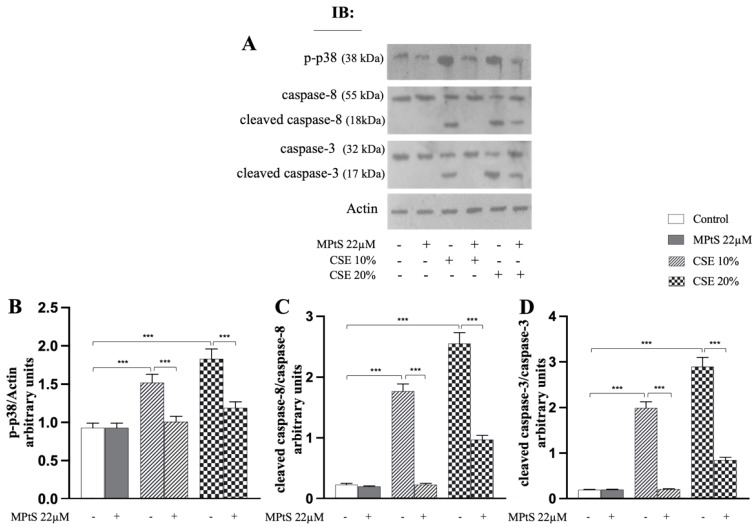
Mixture of plant sterols (MPtS) inhibits caspase 8/caspase 3 cleavage and phosphorylation of p38 MAPK in cigarette smoke extract (CSE)-induced eryptosis. (**A**) Representative images of immunoblotting analysis of p-p38 MAPK, cleaved caspase 8 and cleaved caspase 3 in 10% or 20% CSE-treated RBCs incubated for 4 h in absence or in co-treatment with 22 μM MPtS. Protocol of analysis at Section 4.7. (**B**–**D**) densitometric analysis of p-p38 MAPK, cleaved caspase 8 and cleaved caspase 3 levels normalized for actin. For (**C**,**D**), values are the ratio between cleaved and uncleaved caspases. Values are the means of bands’ densitometry of three separate experiments with comparable results carried out on the same blood sample. *** *p* < 0.001 (ANOVA associated with Tukey’s test).

## Data Availability

All important data is included in the manuscript.

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
