# Peer review of "A Mixture of Dietary Plant Sterols at Nutritional Relevant Serum Concentration Inhibits Extrinsic Pathway of Eryptosis Induced by Cigarette Smoke Extract"

_ijms, 2023, doi:10.3390/ijms24021264_

Round 1

Reviewer 1 Report

This interesting study has focused on a way to decrease the sensitivity of red blood cells to eryptosis induced by cigarette smoke, in vitro. The introduction clearly indicates the mechanism of eryptosis and its increase in smokers, providing a sound rationale for the study.

From a fundamental point of view, eryptosis is reported as an extremely fast way of red blood cell destruction. Why did the authors decide upon a 4 hour incubation? Is the rather low rate of eryptosis, especially by annexin and ceramide staining, potentially related to a weak effect of the cigarette smoke concentrations used? Did the authors, on top of FS data (which should be shown, at least in supplementary) examine whether there was any cell loss during their experiments? Since Ficoll Hypaque gradient centrifugation was used, the suspensions were likely to contain some neutrophils. Similarly, what was the rationale of using a 0.4% hematocrit?

Please define what is intended by “physiological serum concentration”. I guess this is the concentration demonstrated to be reached after consumption of a milk-based preparation of MPts, but this cannot be described as “physiological”. This is better explained line 167 but still should not be referred to as “physiological”

Line 93; what does “reporting values on control levels” mean? In this paragraph use “with” or “after” treatment rather than “in”. For data shown in Figure 3, which criterion was used to position the integration cursor of fluorescence intensity? Changes are rather small and might be better visualized by using an overlay strategy.

Lines 137, 138, there is no need to mention again ref. 4. This whole paragraph needs thorough English editing.

Line 168, “contrasts” is not the appropriate verb. Again this whole paragraph requires editing.

Please use consistent units in the Methods section i.e.L, mL, µL…

Line 244, you did not use fluorescence active cell sorting (FACS) but flow cytometry with no cell recovery. The EPICS XL does not perform sorting. This is a frequent misuse of the acronym FACS!

Minor

The manuscript s globally well written but should be thoroughly checked for grammar and a few typos.

Please define MPts at first occurrence.

Do not use USA when the state of a company’s location is mentioned. Only provide the full information (company, city, state or company, city, country) on the first instance, then use only the company name. Catalog numbers are optional.

Please homogenize the reference style.

Author Response

Ms. Ref. No.:  ijms-2134177

Title: A mixture of dietary plant sterols at physiological serum concentration inhibits extrinsic-pathway of eryptosis induced by cigarette smoke extract.

Point-by-point response:

Mr. Murray Mei (Assistant Editor),

We would like to thank the editor and reviewers for the corrections, comments, and suggestions which have helped us to improve the article.

Reviewer's comments:

Reviewer #1:

  1. From a fundamental point of view, eryptosis is reported as an extremely fast way of red blood cell destruction. Why did the authors decide upon a 4 hour incubation? Is the rather low rate of eryptosis, especially by annexin and ceramide staining, potentially related to a weak effect of the cigarette smoke concentrations used? Did the authors, on top of FS data (which should be shown, at least in supplementary) examine whether there was any cell loss during their experiments? Since Ficoll Hypaque gradient centrifugation was used, the suspensions were likely to contain some neutrophils. Similarly, what was the rationale of using a 0.4% hematocrit?

Response:

  1. We opted for a 4-hour experimental model because in our last work [past ref 10-current ref 13], we saw the activation of the extrinsic way of death in a time course from 0 to 6 hours. We are convinced that the 4-hour model was optimal for the study of a potential inhibitory effect of MPtS in this extrinsic eryptosis induced by CSE as reported in line 101.

  1. We can say with certainty that the relatively “low” levels of fluorescence related to annexin and ceramide are not only related to the low concentration of CSE but to the exposure time. We can say this because preliminary studies carried out to develop the in vitro model of RBC-CSE in the previous study [past ref 10-current ref 13], have shown not only a dose-dependent increase of the eryptosis but also a time-dependent between 24 and 48 hours.

  1. We attach as additional supplementary material the measured FS, together with the information of ROS, GSH and Ca2+ values (Figure S1. Supplementary material). We found that there was no manifest cell loss because we do not appreciate any cell population with extremely lower (debris) o higher (linked with hemolysis) cell size in the cytogram. We found that there was no loss of erythrocytes because we evaluated hemolysis and cell count.

  1. Cytograms of the cell suspensions analyzed did not consider neutrophil populations. For the isolation of erythrocytes, after the application of Ficoll Hypaque gradient centrifugation, the platelets and leukocytes-containing supernatant were discarded and only the pellet containing erythrocytes was used for experiments.

  1. The rationale of using 0.4% hematocrit is that this low hematocrit minimizes mutual erythrocyte interaction as reported by Abed et al. 2017 and is the common concentration of erythrocytes normally used in the eryptosis experiments developed by Lang’s research group (pioneer in this kind of experiments).

Abed et al. 2017.  Stimulation of Erythrocyte Cell Membrane Scrambling by C-Reactive Protein. Cell Physiol Biochem. 2017;41:806-818.

  1. Please define what is intended by “physiological serum concentration”. I guess this is the concentration demonstrated to be reached after consumption of a milk-based preparation of MPts, but this cannot be described as “physiological”. This is better explained line 167 but still should not be referred to as “physiological”

Response:

To better define "physiological serum concentration", we replaced “physiological” with “nutritional relevant” in the title and in the whole manuscript.

  1. Line 93; what does “reporting values on control levels” mean? In this paragraph use “with” or “after” treatment rather than “in”. For data shown in Figure 3, which criterion was used to position the integration cursor of fluorescence intensity? Changes are rather small and might be better visualized by using an overlay strategy.

Response:

  1. In the paragraph “in” is replaced as suggested. For “reporting values on control levels” we mean that co-treatment with CSE and MPtS reports the fluorescence values of the treated RBCs as those of control.

  1. The cursor has been positioned considering the increase in fluorescence intensity from 100 onwards. We agree that the overlap strategy is a good method of representing cytofluorometric data.

In this case, we used fluorescence histograms as a representative image, while differences in fluorescence mean were highlighted more in bar graphs. Therefore, we prefer to maintain the current display of data in figure 3.

  1. Lines 137, 138, there is no need to mention again ref. 4. This whole paragraph needs thorough English editing.

Response:

Ref 4 removed in the paragraph and the sentences were edited.

  1. Line 168, “contrasts” is not the appropriate verb. Again, this whole paragraph requires editing.

Response:

“contrasts” is replaced with “counteracts”, also the paragraph is edited.

  1. Please use consistent units in the Methods section i.e.L, mL, µL…

Response:

Done.

  1. Line 244, you did not use fluorescence active cell sorting (FACS) but flow cytometry with no cell recovery. The EPICS XL does not perform sorting. This is a frequent misuse of the acronym FACS!

Response:

Thanks for the observation. It was a typo error and has been corrected.

  1. Please define MPts at first occurrence.

Response:

In addition to the abstract, all abbreviations were defined in their first appearance in the body of the manuscript.

  1. Do not use USA when the state of a company’s location is mentioned. Only provide the full information (company, city, state or company, city, country) on the first instance, then use only the company name. Catalog numbers are optional.

Response:

Done.

  1. Please homogenize the reference style.

Response:

Done.

Reviewer 2 Report

The manuscript aims to bring data regarding the effects of plant sterols on erythrocytes affected by cigarette smoke extract as well as their mechanisms. I have only minor comments.

Typing errors – 22µM (line 22), eriptotic (lines 50, 171), releave (line 98), 32,13±1,23 (line 219)

All abbreviation should be defined in the main text, despite their definition in the abstract (e.g. RBC, PS, DISC, MPtS...)

In the figure legends, all abbreviations used in the corresponding figures should be explained.

At the beginning of the chapter 2.1, 2.2 and 2.3, there is information that is more suitable for the “Materials and Methods” part.

Line 137-138: I suggest the authors replace such statements either in the introduction or to the discussion section (Activation and signaling of the p38 MAP kinase pathway also appears to be involved in several eryptotic pathways [4].)

The first paragraph of the discussion section is more suitable for introduction – or it is recapitulation of information provided in the introduction, thus, it is useless.

Can the authors provide precise information in the following sentences?

-        line 174-175: “…have not been observed.“ – in the present or which study? intracellular Ca2+ levels and changes in redox balance were not determined in this study…

-        line 175: “…has been observed previously.“ – what exactly has been observed previously?

What was the reason of 4-hour lasting incubation? Please discuss its duration.

Line 187: ASK1-MKK 3/6 axis – please explain what does it mean

Lines 188 – 190: “…PtS like β-sitosterol and ergosterol appear to have modulated activities of p38 MAPK” – it may seem that in erythrocytes… please add in what cell type it was observed.

I suggest the authors briefly discuss the finding that intact RBCs (not exposed to CSE) were unable to profit from the incubation with plants sterols.

Author Response

Ms. Ref. No.:  ijms-2134177

Title: A mixture of dietary plant sterols at physiological serum concentration inhibits extrinsic-pathway of eryptosis induced by cigarette smoke extract.

Point-by-point response:

Mr. Murray Mei (Assistant Editor),

We would like to thank the editor and reviewers for the corrections, comments, and suggestions which have helped us to improve the article.

Reviewer's comments:

Reviewer #2:

  1. Typing errors – 22µM (line 22), eriptotic (lines 50, 171), releave (line 98), 32,13±1,23 (line 219)

Response:

The errors have been fixed.

  1. All abbreviation should be defined in the main text, despite their definition in the abstract (e.g. RBC, PS, DISC, MPtS...)

Response:

In addition to the abstract, all abbreviations were defined in their first appearance in the body of the manuscript.

  1. In the figure legends, all abbreviations used in the corresponding figures should be explained.

Response:

All the abbreviations used in the figures are now explained in the legends.

  1. At the beginning of the chapter 2.1, 2.2 and 2.3, there is information that is more suitable for the “Materials and Methods” part.

Response:

Thanks for the suggestion but we would prefer to leave there the sentences to give a better interpretation of the results and better comprehension for readers.

  1. Line 137-138: I suggest the authors replace such statements either in the introduction or to the discussion section (Activation and signaling of the p38 MAP kinase pathway also appears to be involved in several eryptotic pathways [4].)

Response:

We removed the reference and added “characterized by the phosphorylation of the enzyme” for better introduce the results about p38 MAPK.

  1. The first paragraph of the discussion section is more suitable for introduction – or it is recapitulation of information provided in the introduction, thus, it is useless.

Response:

According to the reviewer’s suggestion the first paragraph of the discussion section has been moved to the introduction. Consequently, all the references have been re-numbered.

  1. Can the authors provide precise information in the following sentences?

-        line 174-175: “…have not been observed.“ – in the present or which study? intracellular Ca2+ levels and changes in redox balance were not determined in this study…

-        line 175: “…has been observed previously.“ – what exactly has been observed previously?

Response:

The reviewer is right. We have already measured FS, ROS, GSH and Ca2+ values but did not provide the information in the manuscript. Now this information is provided as Figure S1. Supplementary material.

  1. What was the reason of 4-hour lasting incubation? Please discuss its duration.

Response:

We opted for a 4-hour experimental model because in our last work [past ref 10-current ref 13], we saw the activation of the extrinsic way of death in a time course from 0 to 6 hours. We are convinced that the 4-hour model was optimal for the study of a potential inhibitory effect of MPtS in in this extrinsic eryptosis induced by CSE as reported in line 101.

  1. Line 187: ASK1-MKK 3/6 axis – please explain what does it mean

Response:

The abbreviation means: Apoptosis Signal-Regulating Kinase 1 (ASK1)- MAPK kinases (MKK) 3/6 axis and has been explained in text.

  1. Lines 188 – 190: “…PtS like β-sitosterol and ergosterol appear to have modulated activities of p38 MAPK” – it may seem that in erythrocytes… please add in what cell type it was observed.

Response:

The information requested by the reviewer on the cell types has been inserted.

  1. I suggest the authors briefly discuss the finding that intact RBCs (not exposed to CSE) were unable to profit from the incubation with plants sterols.

Response:

We do not understand what the reviewer’s mean since intact RBCs only exposed to plant sterols without CSE treatment have no detrimental or injury effect in terms of eryptosis hallmarks as shown in all the results presented in the manuscript. Thus, this is per se a profit/benefit. In fact, in a previous study of our research groups (Álvarez-Sala et al 2018-ref 14 of previous version of the manuscript, current ref 19) it was already shown that plant sterols treatment on RBCs have neutral effects on the eryptotic process, confirming findings of in vivo studies where plant sterols incorporation into the erythrocyte membrane does not alter membrane properties such as rigidity, osmotic fragility, and deformability.

Alvarez-Sala, A.; López-García, G.; Attanzio, A.; Tesoriere, L.; Cilla, A.; Barberá, R.; Alegría, A. Effects of Plant Sterols or β-Cryptoxanthin at Physiological Serum Concentrations on Suicidal Erythrocyte Death. J. Agric. Food Chem. 2018, 66, 1157–1166, doi:10.1021/acs.jafc.7b05575.
